# Clinical Outcome of Two-Stage Revision after Periprosthetic Shoulder Infection

**DOI:** 10.3390/jcm10020218

**Published:** 2021-01-09

**Authors:** Sebastian Klingebiel, Christoph Theil, Georg Gosheger, Kristian Nikolaus Schneider, Thomas Ackmann, Maximilian Timme, Dominik Schorn, Dennis Liem, Carolin Rickert

**Affiliations:** 1Department of Orthopaedics and Tumororthopaedics, University Hospital Muenster, 48149 Münster, Germany; Christoph.Theil@ukmuenster.de (C.T.); Georg.Gosheger@ukmuenster.de (G.G.); kristian.schneider@ukmuenster.de (K.N.S.); Thomas.Ackmann@ukmuenster.de (T.A.); Carolin.Rickert@ukmuenster.de (C.R.); 2Institute for Legal Medicine, University Hospital Muenster, 48149 Münster, Germany; M.timme@uni-muenster.de; 3Department for Shoulder and Elbow Surgery, Paracelsus Clinic Bremen, 28329 Bremen, Germany; schorn.ortho@gmail.com; 4Sporthopaedicum Berlin, 10627 Berlin, Germany; Liem@sporthopaedicum.de

**Keywords:** PJI, total shoulder arthroplasty, reverse shoulder arthroplasty, revision surgery, prosthesis exchange

## Abstract

Background: Periprosthetic shoulder infections are devastating complications after shoulder arthroplasty. A potential treatment concept is a two-stage prosthesis exchange. Data are sparse in terms of clinical outcome, including infection-free survival and patient satisfaction after this procedure. In the present study, we investigated recurrence of infection, revision-free survivorship and clinical outcome following two-stage revision due to periprosthetic shoulder infection. Furthermore, reasons for poor outcome were analyzed. Methods: Sixteen patients undergoing two-stage revision after shoulder joint infection were retrospectively identified. Recurrence of infection was analyzed by Kaplan–Meier survival curve. Clinical outcome was quantified with subjective shoulder value (SSV), “quick” Disabilities of the Arm, Shoulder and Hand (qDASH) and Rowe score. Range of motion (ROM) was measured pre- and postoperatively. Postoperative scores and ROM were compared in a subgroup analysis according to different reimplanted prosthesis types. Results: The reinfection-free implant survival was 81% after one year and at final follow-up (FU; mean of 33.2 months). The overall revision-free survival amounted to 56% after one year and at final FU. Patients who received reverse shoulder arthroplasty (RSA) as part of reimplantation had less disability and long-term complications. This group demonstrated better subjective stability and function compared to patients revised to megaprostheses or large-head hemiarthroplasties. Conclusions: Two-stage revision following periprosthetic joint infection of the shoulder allows appropriate infection control in the majority of patients. However, the overall complications and revision rates due to mechanical failure or reinfection are high. Reimplantation of RSA seem superior to alternative prosthesis models in terms of function and patient satisfaction. Therefore, bone-saving surgery and reconstruction of the glenoid may increase the likelihood of reimplantation of RSA and potentially improve outcome in the case of infection-related two-stage revision of the shoulder.

## 1. Introduction

Primary shoulder arthroplasty (SA) is an expanding field in treating proximal humeral fractures and degenerative shoulder disorders, including osteoarthritis and cuff tear arthropathy (CTA) [1]. In 2012, 34 glenohumeral joint replacements per 10^5^ inhabitants were performed in Germany, which means a tripling of the annual incidence in the past decade [2]. With an increasing number of shoulder replacement surgeries, orthopedic surgeons are increasingly confronted with complications like implant failure [3]. Apart from mechanical disorders, periprosthetic joint infections (PJIs) are serious complications associated with prolonged hospitalization and long-term functional constraints [4]. Several influencing risk factors, such as bone and soft tissue status, comorbidities, socioeconomic status, age and gender, that may predispose for PJI are discussed [5]. However, male gender and young age at primary implantation were identified as the most important influencing factors [6,7]. It is reported that the incidence of PJI following primary SA ranges from 0 to 4% [8,9]. Distinct guidelines for the clinical management are missing and appropriate treatment concepts are heterogeneous [10,11,12,13]. Treatment ranges from exclusive antibiotic treatment to surgical revision with irrigation and debridement or staged prosthesis exchange up to definitive resection arthroplasty [14,15,16], whereas most studies recommend removal of the implant for chronic infections [8,17,18]. Infection control can also be achieved by resection arthroplasty [13,19,20,21] or permanent spacer retention [22]. However, both procedures have a poor functional outcome with a high rate of dissatisfied patients [13,19,22]. Therefore, the most established procedures are the one- and two-stage prosthesis exchange utilizing similar approaches as for hip and knee PJI [22,23,24,25].

A clear superiority of one procedure over the other in terms of infection control and functional outcome has not been identified to date [17]. However, several authors consider a two-stage approach the gold standard, as it is not limited by previous identification of the causative organism or soft tissue conditions and is associated with a potentially better infection control [18]. Usually for a two-stage approach, the infected prosthesis is first removed completely and a temporary bone cement spacer is implanted [17]. Reimplantation of a new prosthesis is performed several weeks later, provided that the infection appears to have been completely eliminated. However, two-staged exchange is associated with the morbidity of a second surgery and recurrence of infection varies; it may still be high according to Strickland et al. [26], who report a reinfection rate of up to 50%. Furthermore, there is a lack of studies investigating functional outcome and patient satisfaction after two-stage exchange due to PJI. Therefore, in the present study, infection control, revision-free survival and functional outcome after this procedure were studied and reasons for poor outcome and reduced patient satisfaction were analyzed.

## 2. Patients, Materials and Methods

### 2.1. Inclusion Criteria and Patients

In this single-center study, we retrospectively analyzed the consecutive patients undergoing revision shoulder arthroplasty performed in our department from November 2010 to November 2019. We identified 122 patients. Of these, 106 were aseptic revisions (one-stage exchange, prosthesis conversions, component changes) and 16 (6 male, 10 female) underwent a two-stage exchange for PJI. Mean age at time of reimplantation was 65 years (range 41–76). Further demographic details are presented in Table 1. The exclusion criterion were revision of SA for other cause than infection, and repeat infections after already performing two-stage exchange following PJI were not considered new cases.

The study was approved by the local ethics committee (Westfälische Wilhelms-Universität Münster (WWU), ref. no. 2020-496-f-S).

PJI comprised low-grade (symptoms > 3 months, chronical pain, joint fistula, increased infection parameters) and acute infections (symptoms for hours or a few days, systemic sepsis). The cases between the two definitions with symptom duration between 7 days and 3 months were referred to internally as “sub-acute”. However, these cases were statistically combined with the acute cases.

When diagnosing patients, a standardized algorithm was followed, irrespective of the presentation (acute infections vs. elective in low-grade infections). This algorithm always included a clinical examination, a laboratory analysis and, if the suspicion of infection was confirmed, joint aspiration. For selected cases with high suspicion, but a “dry tap”, or clotted specimens, an open biopsy was performed. In detail, diagnosis was based on clinical symptoms, such as local warmth, stiffness, intermittent joint swelling or fistula in combination with elevated infection parameters, especially serum C-reactive protein (CRP) (>0.5 mg/dL) and elevated leucocytes in blood count. Further criteria were positive aspiration with elevated leucocytes (total leucocyte count >3000 and percentage of neutrophils >65%) and positive aspiration culture. Additionally, the majority of patients’ positive microbiological findings (culture in open biopsy, intraoperative cultures) or macroscopic intraoperative findings like pus could confirm the suspected PJI.

The mean follow-up (FU) amounted to 32 months (range 12–85). According to Xu et al. [27], a 12-month minimum follow-up period is sufficient to investigate treatment failure in PJI, therefore, the minimum follow-up in the present study was 12 months. Range of motion (ROM) was measured prior to two-stage revision and at last FU. Postoperative outcome was measured with subjective shoulder value (SSV) which, according to Gilbart et al. [28], shows a high correlation to the constant score (CS). The “quick” Disabilities of the Arm, Shoulder and Hand (qDASH) and Rowe score were also measured [29,30]. The postoperative scores and ROM were compared in a subgroup analysis (reverse shoulder arthroplasty (RSA) vs. revision prostheses (hemiarthroplasty (HA)/reverse proximal humeral replacement (RPHR)) and all reverse prostheses vs. HA with CTA head).

### 2.2. Treatment of PJI by Two-Stage Exchange

The first step of two-stage revision included complete removal of implant components and implantation of an antibiotic-impregnated bone cement spacer (polymethylmethacrylate—PMMA) (Figure 1). After surgery, antibiotic treatment was initiated using broad-spectrum antibiotics. The intravenous antibiotic treatment was carried out for at least two weeks with subsequent oral application for a further 4 weeks. During the interval until prosthesis reimplantation, the shoulder was immobilized in a brace. Reimplantation was indicated in the presence of normal CRP and the absence of clinical signs of infection. After reimplantation (Figure 2), the shoulder was again immobilized for 4 weeks. On day one, rehabilitation was started with passive movement exercises under physiotherapeutic supervision. After seven to fourteen days, active assisted exercises were allowed according to intraoperative joint stability. If the same microorganism was detected during reimplantation as during explantation, or a new microorganism was detected in more than one of 5 samples during reimplantation, an antibiotic treatment was administered for 6 weeks.

### 2.3. Statistical Analysis

All data were collected using a Microsoft Office Excel sheet (Microsoft Corp, Seattle, WA, USA) and all analyses were performed using SPSS for Windows Version 26 (IBM Corp, Redmont, VA, USA). We analyzed data distribution using the Kolmogorov–Smirnov test and present mean/medians with corresponding ranges or 25–75% interquartile ranges depending on data normality. Groupwise comparisons were done using the chi-squared test for cross tables and the Mann–Whitney U test or Student’s *t*-test for (non-)parametric testing. Implant survivorship was calculated using the Kaplan–Meier method and survival curves for the primary endpoints (reinfection, overall revision) were generated. A *p*-value of <0.05 was considered significant.

## 3. Results

### 3.1. General

We identified 12 low-grade PJIs and four acute infections. CRP was elevated in 12 patients with a mean of 3.0 mg/dL (range < 0.5–15.3; IQR 0.7–4.3). In four cases, joint fistulas indicated chronical PJI. The median retention period of the PMMA spacers before prosthesis reimplantation was 18.7 weeks (IQR 10–15.8). Further baseline data are summarized in Table 1.

The median CRP level before reimplantation was 0.9 mg/dL (IQR 0.5–1.1). In 12 patients it was below 1 mg/dL, and in four patients it ranged between 1.1 and 2.9 mg/dL. Nine of 16 patients (56%) had RSA reimplanted. In three patients, glenoidal augmentation was performed with allogenic bone material to restore glenoid defects. Four patients (25%) were revised to reverse proximal humeral replacement due to humeral bone loss (Table 2). Three patients (18%) received a large-head HA (Figure 3). HA was reimplanted as a result of severe glenoidal bone defects based on the classification proposed by Antuna (Table 2). Average hospital stay after prosthesis reimplantation was 10 day (range 6–14). Three patients died unrelated to PJI (1 × poor general condition due to multimorbidity, 1 × stroke, 1 × liver failure).

### 3.2. Recurrence of Infection and Revision-Free Implant Survivorship

The mean period from index prosthesis implantation until diagnosis of acute or chronic PJI followed by prosthesis explantation and spacer interposition amounted to 14.2 months (IQR 3.2–28.4; range 1.5–34). The infection recurred in 19% (3/16) of patients. One patient who had undergone reverse proximal humeral replacement (Figure 4) had acute early-onset PJI after 1.5 months, while two patients (1 RPHR, 1 RSA) showed late-onset and low-grade reinfection after seven and twelve months, respectively. These patients underwent repeat revision surgery, with two patients undergoing two-stage exchange again and the other undergoing definitive resection arthroplasty. The latter patient was subsequently free of pain in the mid-term with significantly reduced function, but has now been free of infection for over 4 years.

The reinfection-free implant survivorship was 81% (95% CI (62–100%)) after one year and at final FU of 33.2 months (IQR 13.25–59.75, range 1.5–87.0) (Figure 5). The overall revision-free survival (revision for any cause) amounted to 56% (95% CI 32–90%) after one year and at final follow-up. Thirty-one percent (5/16) of patients underwent revision surgery following non-infective complications (Table 3). Two patients with dislocations underwent inlay exchange and did not suffer from further instability, while, on the other hand, two patients with RPHR had recurrent instability despite inlay revision and lengthening of the modular reverse proximal humeral replacement. One was converted to anatomic modular megaprosthesis and one remained chronically unstable. Another patient with chronic instability after reimplantation of large-head HA refused repeat revision surgery. With the numbers available, there were no differences in implant survival with respect to the type of prosthesis used during reimplantation surgery. However, there was a 75% (3/4) revision rate for infection and non-infection reasons in RPHR while only 11% (1/9) of patients who underwent reconstruction with a non-megaprosthetic RSA suffered reinfection and 33% (3/9) underwent overall revision surgery.

### 3.3. Microbiological Findings

Sixty-nine percent (11/16) were culture-positive PJI. Eighteen percent (2/11) of these were polymicrobial. In 31% (5/16), it was a culture-negative infection (Table 3). Fifty-three percent (30/57) of the microbiological samples taken indicated positive germ detection. *Cutibacterium acnes* was found in 25% (4/16). Multisensitive and resistant staphylococci as well as intestinal bacteria (*E. coli*, *E. faecalis*) were also found, Table 2. During reimplantation, positive microbiological samples were detected in 50% (8/16) of the patients. *Methicillin-resistant Staphylococcus epidermidis* (MRSE) was found in 75% (6/8). In two patients, a microbiological congruence of the positive culture was found between explantation and reimplantation, and in both cases *MRSE* was identified. In another patient, new *MRSE* was detected in 3/5 samples. For these three cases, prolonged antibiotic therapy was conducted.

### 3.4. Functional Outcome

Overall postoperative patient satisfaction was low with a subjective shoulder value (SSV) at last FU of 56/100 (IQR 28–78), qDASH with a mean of 40/100 (range 11–83) and a mean Rowe score of 52/100 (range 20–85) (Table 4). However, patients who underwent RSA showed significantly better postoperative SSV compared to RPHR and prostheses with large-head HA (72/100 vs. 29/100, *p* = 0.001) (Table 5). Comparably, the Rowe score was significantly higher and qDASH significantly lower in patients with RSA (qDASH: RSA: 26 (IQR 17–41) vs. RPHR/CTA: 58 (IQR 41–81), *p* = 0.003; Rowe: RSA: 67 (48–57) vs. RPHR/CTA: 33 (IQR 20–55), 0.003). Patients with conventional RSA had less disability and showed better subjective stability compared to the megaprostheses and HA.

Patients who underwent SA for proximal humerus fracture had a trend for a lower Rowe score after two-stage exchange (35 (IQR 20–65) vs. 73 (51–85), *p* = 0.056) while the qDASH and SSV were not different (*p* = 0.43 and *p* = 0.18, respectively).

Furthermore, patients who were reconstructed with RSA (*n* = 9) displayed a better range of motion for internal rotation (50° vs. 29°, *p* = 0.031), abduction (79° vs. 29°, *p* = 0.002) and flexion (82° vs. 30°, *p* = 0.001) compared to megaprostheses and large-head HA (Table 6). Again, patients with SA for proximal humerus fracture showed worse postoperative range of motion compared to other indications (abduction: 45 (IQR 20–80) vs. 80 (IQR 80–90), *p* = 0.073 and flexion: 55 (20–80) vs. 85 (IQR 80–90), *p* = 0.042).

## 4. Discussion

The objectives of this study were the analysis of the recurrence of infection and infection-free survival after two-stage exchange of shoulder prosthesis following PJI. The most important finding was that infection control was achieved in the majority of patients, at over 80%. However, overall revision-free prosthesis survival was 56% and therefore major complications leading to revision surgery were common after two-stage treatment. If the anatomical conditions, particularly on the glenoid side, allowed reimplantation of an RSA, there was a trend towards increased patient satisfaction and improved postoperative function than after treatment with alternative prostheses.

### 4.1. Infection Control after Two-Stage Revision

The primary objective when performing revision surgery for infection is infection control [31,32,33], with most comprehensive data available for the two-stage approach [8,17,18,26]. In a recent review from 2019, Kunutsor et al. [17] included 27 studies with 351 patients. The recurrence rate of infection was 16.2% after a median FU of 3.9 years. Therefore, our results are in line with the literature. In contrast, eight studies of a one-stage prosthesis exchange with a FU of 3 years were reviewed and it is reported that 8% of the included patients (12 of 147) presented recurrent infection. According to these data, the less common one-stage treatment seems equivalent or even beneficial in terms of infection control. However, especially in the case of preoperatively unidentified microorganisms, a two-stage approach is considered as the method of choice by several authors to eradicate biofilm-forming bacteria, as local targeted antibiotic treatment is crucial [31,34]. Appropriate surgical restoration with extensive soft tissue and bone debridement appears to be crucial for adequate infection control [18]. In this context, an advantage of two-stage revision is the opportunity of a further tissue debridement without the need to perform reconstruction right away [5]. Considering that PJI after SA is still a rare but growing complication, future comparative studies should evaluate the ideal treatment approach.

The fact that *MRSE* was detected in almost 50% of cases (7/16; Table 3) at the time of the second stage of reimplantation is remarkable. For our part, however, only those cases in which *MRSE* had already been detected at the time of the first step (explantation) were evaluated as genuine infections, as the risk of contamination is present. Two of these patients in our study underwent revision for reinfection, however. Future studies should focus on optimizing antibiotic treatment for these bacteria as they might persist despite radical debridement.

### 4.2. Complication Rate after One- and Two-Stage Revision

The revision rate of 31% in our cohort for non-infectious complications is consistent with reports from the literature: Strickland et al. [26] reported up to 74% overall postoperative complications and a 29% revision rate following reimplantation in 17 patients (19 shoulders) [18,26]. Frequent complications were joint dislocation, chronic instability, acromion fracture, superficial wound healing disorders or hematoma formation. It has been reported that the complication rate may be lower with a single-stage procedure [17]. It may be discussed whether secondary damage caused by the spacer in the two-stage approach is a cause of this. Nevertheless, only one patient of the present study showed anterior spacer dislocation with resulting glenoid erosion (Figure 6). Moreover, the increasing contracture of the surrounding soft tissue caused by the spacer remaining for a longer time may lead to higher risk of postoperative instability.

However, successful stabilization was achieved in all patients with postoperative dislocation following reimplantation of RSA. Our data show that reimplantation of alternative prostheses (large-head HA, proximal humerus replacement) may lead to more chronic instability in the long term. However, glenoid bone stock appears to be a limiting factor for conventional RSA. Therefore, the role of glenoidal augmentation is important in this approach. Nonetheless, future studies are needed on the use of bone grafting for glenoid bone defects in the setting of infection [35].

### 4.3. Clinical Outcome and Postoperative Function after Surgical Treatment of PJI

While pain reduction seems to be achievable in more than half of patients by resection arthroplasty, the functional outcome is poor due to severe limitations of movement [19,20]. Patient satisfaction is low, as has been reported by Braman et al. [19] and Romano et al. [13], who reported a postoperative CS of 30 and 32, respectively. Postoperative abduction and external rotation were described as up to 28° and 8°, respectively [36]. However, these results appear to be poor and somewhat controversial regarding the results reported by Sperling et al. [21], who reported an abduction of 70° and an external rotation of 31°. Considering the serious functional disadvantages, the indication for resection arthroplasty should be reserved for patients with high surgical risk.

For the single-stage approach, Ince et al. [37] reported a poor mean postoperative CS of 33 (*n* = 9). Coste et al. [36] reported better results, presenting a postoperative CS of 66 (*n* = 3). Cuff et al. [10] reported an improvement of mean abduction from 36° to 76° (*n* = 10) using a single-stage exchange. Mean forward flexion was improved from 43° to 80° and external rotation from 10° to 25° [10]. Cuff et al. [10] found no difference in the outcome between the one- and the two-stage approach. A multi-center study by Amaravathi et al. [38] in 2012 included 24 patients, with 12 patients each treated with one- or two-stage prosthesis exchange. The results were largely similar, with improvements in postoperative CS following the one-stage procedure (53 vs. 43). In their systematic review, Kunutsor et al. [17] found no difference between one- and two-staged revision concerning CS (32 vs. 29), American Shoulder and Elbow Surgeons Shoulder Score (ASES) (34 vs. 32), forward flexion (57° vs. 58°), abduction (46° vs. 51°) and external rotation (23° vs. 14°).

The functional results following reconstruction with RSA are consistent with those in the literature. Nevertheless, patient satisfaction and functional outcome in our patients were significantly higher after RSA compared to revision prostheses (RPHR + large-head HA). One reason might be the reduced functionality of large-head HA and modular megaprosthesis. Therefore, the outcome seems to be closely associated with the anatomical conditions and extent of bone and soft tissue loss caused by previous interventions. Correspondingly, patients show a trend towards poorer function and reduced satisfaction after prosthetic treatment due to a humerus fracture.

### 4.4. Strength and Limitations

We are providing data in a little-explored field, considering that current reviews include only about 350 patients. Despite this large amount of data, we were ultimately only able to accrue 16 cases. A subgroup analysis of different reimplanted prostheses was carried out and has displayed clear differences in function and survival, although the numbers available are too small for a meaningful statistical analysis. The shortcomings of this study are the retrospective design and the small number of patients in an inhomogeneous population with multiple influencing factors. Therefore, significance and risk analyses must be viewed with reservation due to the reduced statistical power.

## 5. Conclusions

Two-stage revisions following PJI of the shoulder allow infection control in the majority of patients. However, the overall levels of complications and revision rates due to mechanical failure are high. Reverse shoulder prostheses seem to be superior to alternative models regarding postoperative function and patient satisfaction. Therefore, bone-saving surgery and reconstruction of the glenoid may increase the likelihood of reimplantation of RSA and potentially improve outcome in the case of infection-related two-stage revision of the shoulder. Positive cultures at reimplantation appear to be an issue.

## Figures and Tables

**Figure 1 jcm-10-00218-f001:**
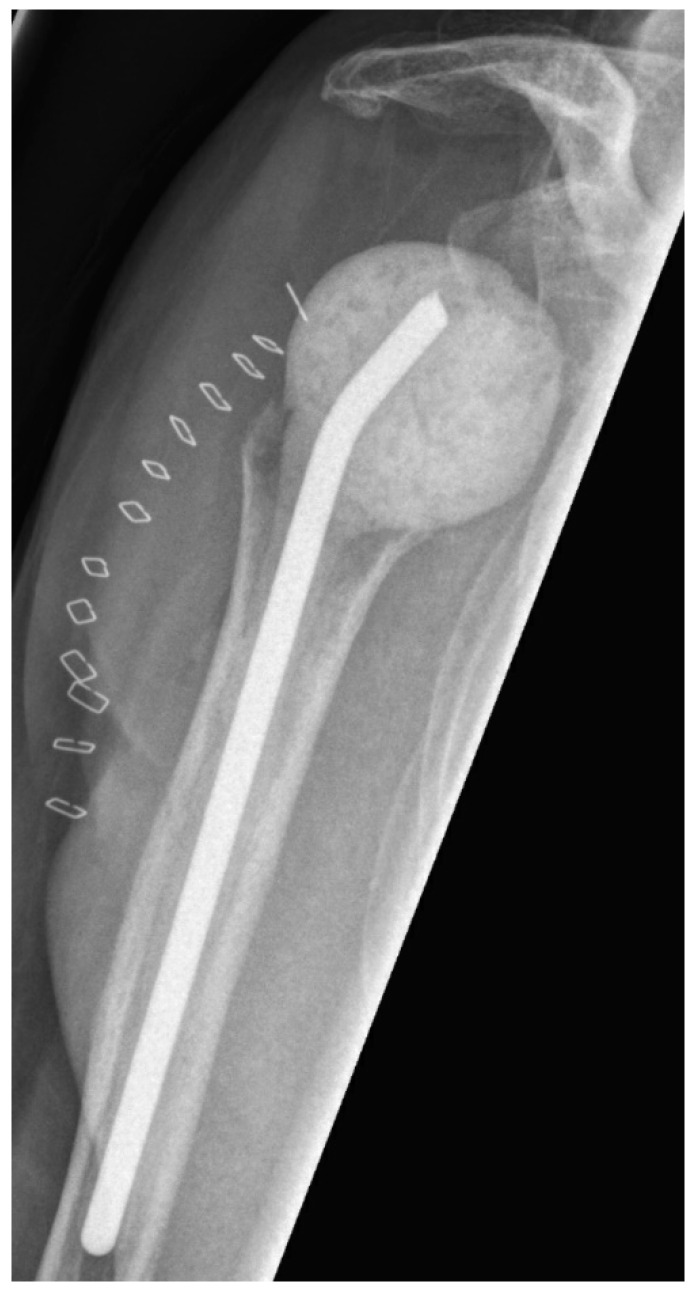
Postoperative anterior–posterior X-ray imaging of a right shoulder after prosthesis explantation and consecutive spacer implantation.

**Figure 2 jcm-10-00218-f002:**
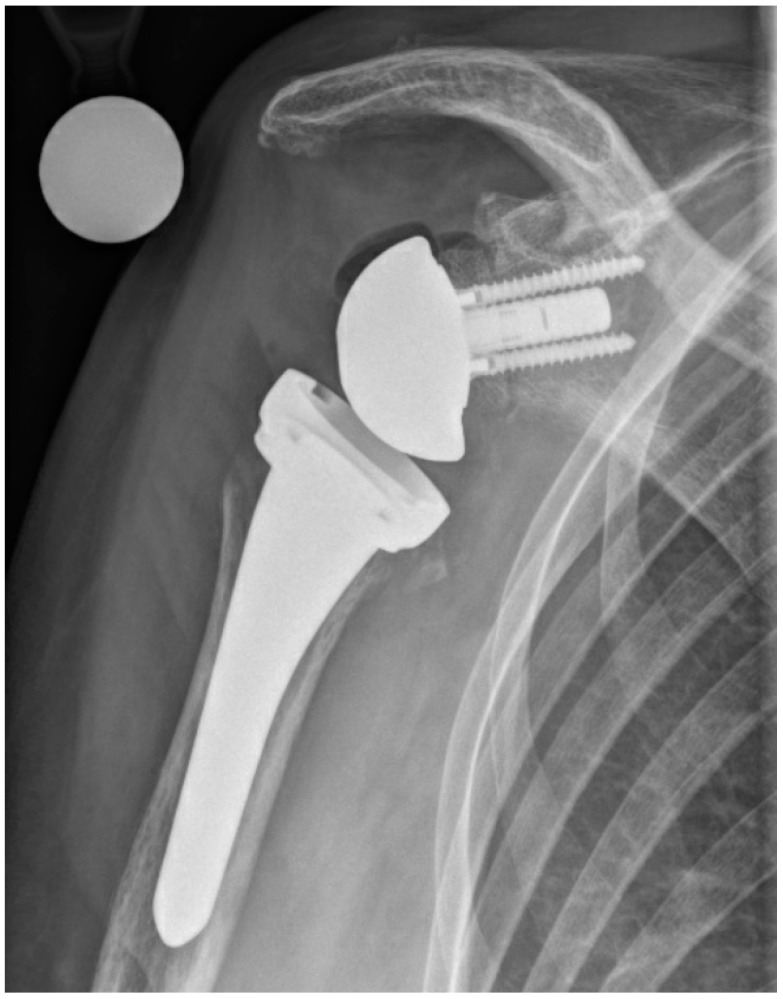
Postoperative anterior–posterior X-ray imaging of a right shoulder after prosthesis reimplantation of a reverse shoulder arthroplasty (uncemented standard stem) and glenoidal augmentation with human allograft due to glenoid erosion.

**Figure 3 jcm-10-00218-f003:**
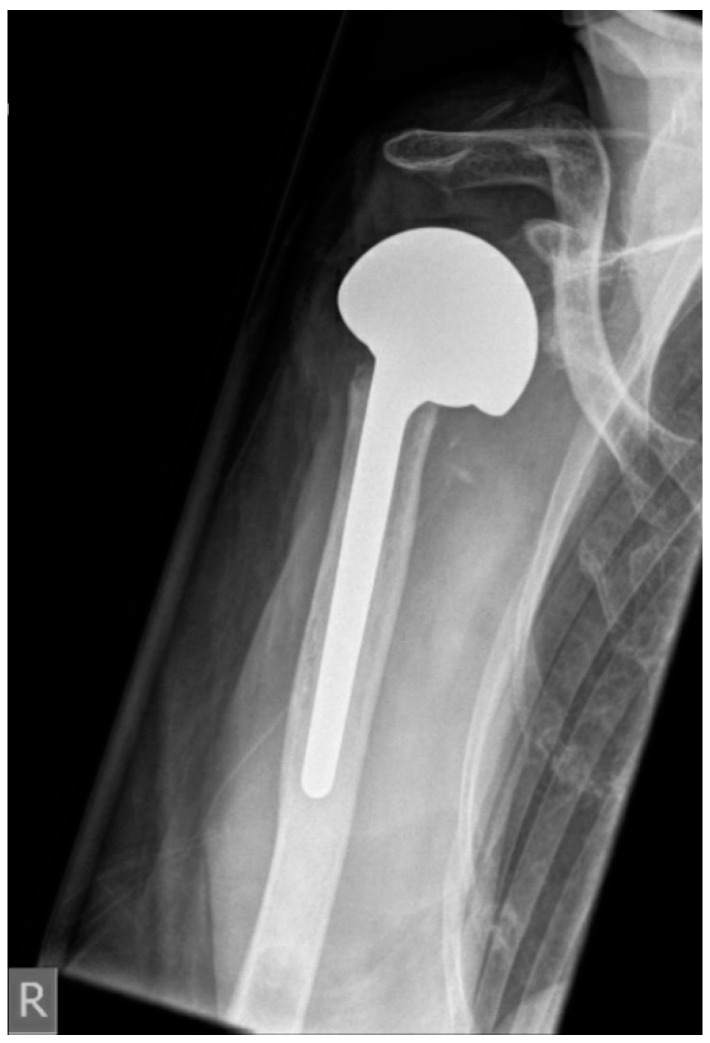
Postoperative anterior–posterior X-ray imaging of a right shoulder after prosthesis reimplantation of hemiarthroplasty with large head (CTA head) due to severe glenoid defect.

**Figure 4 jcm-10-00218-f004:**
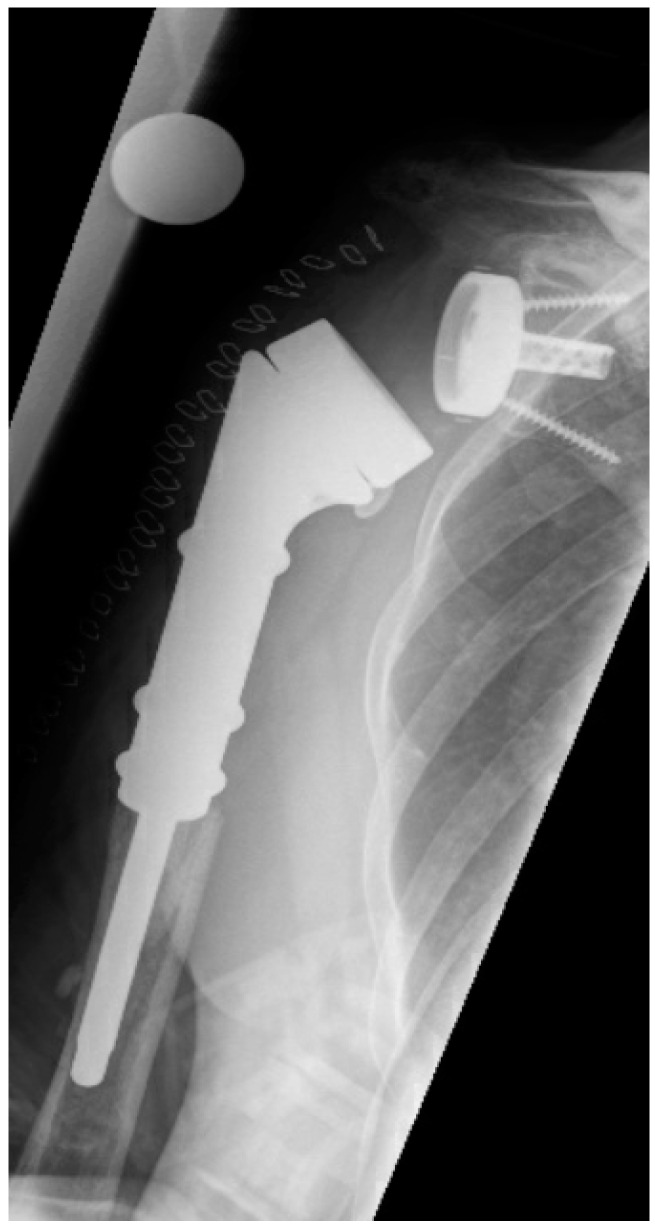
Postoperative anterior–posterior X-ray imaging of a right shoulder after prosthesis reimplantation of reverse proximal humeral replacement due to severe humeral defect.

**Figure 5 jcm-10-00218-f005:**
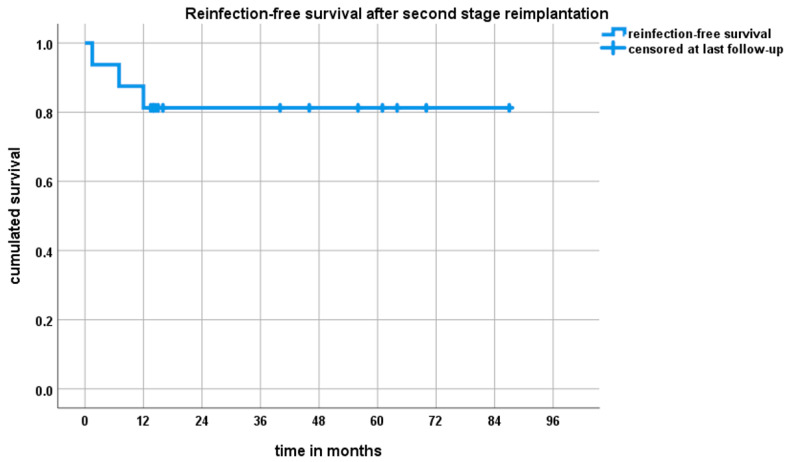
Kaplan–Meier survival curve after two-stage prosthesis exchange following periprosthetic shoulder infection. Three patients showed recurrence of joint infection within 12 months. After 12 months, there were no further reinfections.

**Figure 6 jcm-10-00218-f006:**
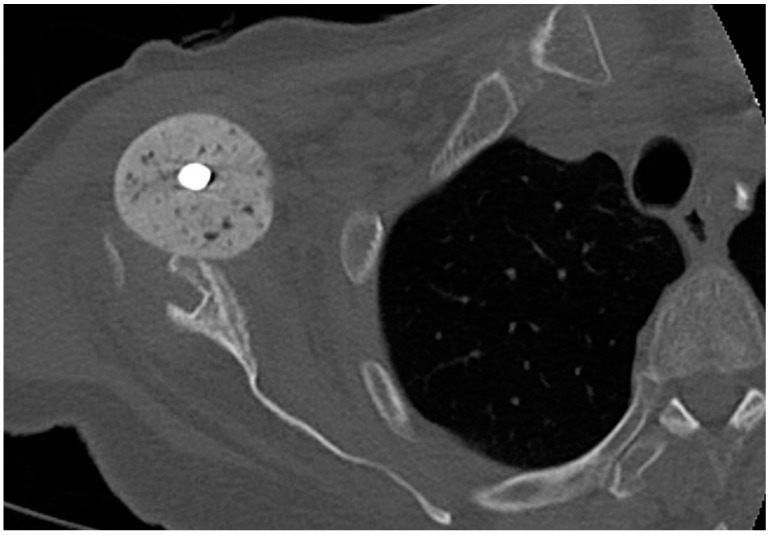
Axial plane of a computer tomography, right shoulder with anterior spacer dislocation and ventral glenoid erosion.

**Table 1 jcm-10-00218-t001:** Baseline table of included patients. BMI: body mass index, SA: shoulder arthroplasty, RSA: reverse shoulder arthroplasty, HA: hemiarthroplasty, CTA: cuff tear arthropathy.

Characteristic	Total (Range)
Patients (*n*)	16
Male (*n*)	6
Female (*n*)	10
Age at reimplantation (y)	65 (41–77)
Height (cm)	170 (154–189)
Weight (kg)	87 (55–120)
BMI (kg/m²)	30 (20–47)
Smokers (*n*)	3
Non-smokers (*n*)	13
Side	
Right (*n*)	9
Left (*n*)	7
Previous operations of affected joint (*n*)	3.75 (2–6)
Initial indication for SA	
Hemiprothesis after fracture (*n*)	7
RSA after fracture (*n*)	3
Primary omarthritis (*n*)	1
Secondary omarthritis (*n*)	1
Cuff tear arthropathy (*n*)	4
Reimplanted prostheses	
RSA (*n*)	9 (56%)
HA with CTA head (*n*)	3 (19%)
Reverse proximal humeral replacement (*n*)	4 (25%)
Hospital stay explantation (d)	14 (7–18)
Hospital stay reimplantation (d)	10 (6–14)
Operating time explantation	124 (95–265)
Operating time reimplantation (min)	117 (83–260)
Explanted stem cemented (*n*)	7
Explanted stem uncemented (*n*)	9
Cemented stem for replantation (*n*)	10
Uncemented stem for replantation (*n*)	6

**Table 2 jcm-10-00218-t002:** Bone loss at second-stage reimplantation.

Bone Loss	*n* (%)
Glenoid	
Mild	7 (44)
Moderate	5 (31)
Severe	4 (25)
Humerus	
Greater tubercle present	5 (31)
Greater tubercle eroded	7 (44)
Proximal humerus defect	5 (31)

**Table 3 jcm-10-00218-t003:** Overview of the included patients with microbial spectrum, replanted prosthesis model and revision, as well as surgical and general complications. *C. acnes*: *Cutibacterium acnes*, *MRSE*: *Methicillin-resistant Staphylococcus epidermidis*, *MSSE*: *Methicillin-sensitive Staphylococcus epidermidis*, *E. facalis*: *Enterococcus faecalis*, *E. coli*: *Escherichia coli*, *S. aureus*: *Staphylococcus aureus*, *S. capitis*: *Staphylococcus capitis*, RSA: reverse shoulder arthroplasty, CTA HA: large-head hemiarthroplasty, RPHR: reverse proximal humerus replacement.

Patient	Culture (Explantation)	Culture (Replantation)	Replanted Prosthesis	Reinfection	Revision	Complication
#1	Negative	Negative	RSA	No	No	-
#2	*C. acnes, MRSE, MSSE*	Negative	RPHR	No	Yes	Acute and chronic instability
#3	Negative	Negative	RPHR	Yes	Multiple	Acute and chronic instability, reinfection, chronic pain
#4	Negative	Negative	RSA	No	No	-
#5	*E. faecalis*	Negative	RSA	No	No	-
#6	Negative	Negative	RSA	No	No	-
#7	*E. coli*	*Corynebacterium*	CTA HA	No	No	-
#8	Negative	Negative	CTA HA	No	Yes	Wound infection, acromion fracture
#9	*MRSE*	*MRSE*	RPHR	Yes	Yes	Reinfection and resection arthroplasty
#10	*S. aureus*	*MRSE*	CTA HA	No	No	Chronic instability
#11	*C. acnes*	*MRSE*	RPHR	No	No	Chronic pain
#12	*C. acnes*	*MRSE*	RSA	No	No	Epileptic seizure
#13	*S. aureus*	*MRSE*	RSA	No	Yes	Single postoperative dislocation
#14	*E. faecalis, MRSE*	*MRSE, Paenibacillus*	RSA	Yes	Yes	Reinfection
#15	*S. capitis*	*MRSE (3/5*)	RSA	No	No	-
#16	*C. acnes*	Negative	RSA	No	Yes	Single postoperative dislocation

**Table 4 jcm-10-00218-t004:** Postoperative scores after second step of prosthesis reimplantation and range of motion (ROM) prior to prosthesis explantation (pre) and after second step of prosthesis reimplantation (post) of all patients. IQR: interquartile range, ER: external rotation, IR: internal rotation, ABD: abduction, FLEX: forward flexion, SSV: subjective shoulder value, qDASH: “quick” Disabilities of the Arm, Shoulder and Hand.

All Patients	Mean (Range)	IQR
SSV (x/100)	40 (11–81)	17–62
qDASH	52 (20–85)	26–80
ROWE (x/100)	54 (10–90)	28–78
Range of motion		
ER pre	7 (0–20)	0–20
ER post	10 (0–50)	0–20
IR pre	39 (0–60)	23–58
IR post	41 (0–80)	23–58
ABD pre	53 (20–90)	23–88
ABD post	57 (0–90)	23–80
FLEX pre	58 (20–130)	33–88
FLEX post	60 (0–90)	25–88

**Table 5 jcm-10-00218-t005:** Postoperative scores after second step of prosthesis reimplantation and range of motion (ROM) prior to prosthesis explantation (pre) and after second step prosthesis reimplantation (post) of patients revised to reverse shoulder arthroplasty (RSA). Values marked in bold show a statistically significant superiority of RSA over the large-head HA and megaprostheses. Empty cells mean lack of significance. IQR: interquartile range, ER: external rotation, IR: internal rotation, ABD: abduction, FLEX: forward flexion, HA CTA: hemiarthroplasty with CTA head (large head), SSV: subjective shoulder value, qDASH: “quick” Disabilities of the Arm, Shoulder and Hand.

RSA	Mean (Range)	IQR	*p*
SSV (x/100)	**72 (50–90)**	**62–85**	**0.003**
qDASH	**26 (35–85)**	**17–42**	**0.003**
ROWE (x/100)	**67 (35–85)**	**48–5**	**0.001**
Range of motion			
ER pre	6 (0–20)	0–15	
ER post	14 (0–50)	0–25	
IR pre	41 (0–60)	25–60	
IR post	**50 (20–80)**	**35–60**	**0.031**
ABD pre	57 (20–90)	20–90	
ABD post	**79 (50–90)**	**75–90**	**0.002**
FLEX pre	63 (20–130)	35–85	
FLEX post	**82 (70–90)**	**35–90**	**0.001**

**Table 6 jcm-10-00218-t006:** Postoperative scores after second step of prosthesis reimplantation and range of motion (ROM) prior to prosthesis explantation (pre) and after second step prosthesis reimplantation (post) of patients revised to reverse shoulder arthroplasty (RSA). IQR: interquartile range, ER: external rotation, IR: internal rotation, ABD: abduction, FLEX: forward flexion, SSV: subjective shoulder value, qDASH: “quick” Disabilities of the Arm, Shoulder and Hand, CTA: cuff tear arthropathy, RPHR: reverse proximal humeral replacement.

CTA + RPHR	Mean (Range)	IQR
SSV (x/100)	29 (10–60)	10–50
qDASH	58 (20– 55)	41–81
ROWE (x/100)	33 (20–60)	20–55
Range of motion		
ER pre	9 (0–20)	0–20
ER post	3 (0–20)	0–0
IR pre	37 (20–50)	20–50
IR post	29 (0–45)	20–45
ABD pre	48 (20–90)	30–75
ABD post	29 (0–80)	10–45
FLEX pre	51 (20–90)	30–90
FLEX post	30 (0–80)	10–40

## Data Availability

The data presented in this study are available on request from the corresponding author. The data are not publicly available due to hospital internal data.

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
