# Peer review of "Clinical Outcome of Two-Stage Revision after Periprosthetic Shoulder Infection"

_jcm, 2021, doi:10.3390/jcm10020218_

Round 1

Reviewer 1 Report

The paper by Klingebiel et al. deals with the clinical evaluation of a two-stage approach to revise infected joint prostheses. The study is retrospective and limited to few hundreds of patients, with different clinical history and different implanted prostheses. These factors significantly reduce the relevance of the information provided by the collected data, as correctly stated by the same Authors at lines 411-417. However, the Authors succeeded in confirming literature trends, thus offering supporting evidence for the suitability of two-stage revision arthroplasty in specific scenarios. I suggest a careful revision of the paper to guarantee that each of the inserted figures is cited in the manuscript and relevant for the discussion (e.g. figure 4 is not recalled in the text). Please also revise English language throughout the manuscript.

Author Response

Thank you and kind regards,

Sebastian Klingebiel

Reviewer 2 Report

Dear sir, the work addressesa interesting topic, but the work presents some important drawbacs. First of all, it is a too long text, it should be shortened. Information on the previous situation of patients is scare. They should clarify how many had been surgical treated prior of arthroplasty. Most were fractures, many had received a previous osteosynthesis?.The classification of the infection between low-grade and acute does not seem appropiate to me (In the results, low-grade, acute, subacute...?). Likewise, the diagnostic process is not well clarified. Perhaps as it is a retrospective study, it may cause errors. There are many other aspects that must be discussed and clarified. For example, half of patients had positive cultures at the time of reimplantation. We must consider that the infection has been eradicated?

Author Response

(The authors gave the same response as above.)

Round 2

Reviewer 2 Report

Dear sir, Thank you for taking my opinion into account, the work has improved and i think  it may be of intertest to readers.